

# DECAF: deep capsule attention network for intelligent auto loan default risk detection

Xiaohui Zang[1,2], Raja Nazim Abdullah[2], Jianhua Tan[3], Minglin Wu[4], Bei Li[1], Zhiyong Chen[1], Enzhou Zhu[1] and Yuan Lei[5]

[1] School of Automotive Engineering, Liuzhou Polytechnic University, Liuzhou, China
[2] Faculty of Management and Economics, Universiti Pendidikan Sultan Idris, Perak, Tanjong Malim, Malaysia
[3] International Center for Education and Exchange, Guangxi Arts University, Nanning, China
[4] School of Mechanical and Automotive Engineering, Guangxi University of Science and Technology, Liuzhou, China
[5] KiwiBridge Ascend Education Limited, Auckland City, New Zealand

Corresponding authors
Raja Nazim Abdullah,
rajanazim@fpe.upsi.edu.my
Minglin Wu,
gxutwuminglin_523@163.com

## ABSTRACT

With the increasing challenges of default risk in the auto loan market, traditional risk assessment methods show significant limitations in coping with the rapidly changing market environment. This study proposes an innovative deep learning architecture—Deep Capsule Attention Network (DECAF)—which integrates capsule networks with self-attention mechanisms, effectively enhancing the intelligent detection capability for auto loan default risks. Experimental evaluations of the model's performance in both conventional and high-risk scenarios demonstrate that the DECAF model achieved an area under the curve (AUC) of 0.924 in the conventional scenario and maintained a high performance of 0.850 in high-risk scenarios. Additionally, DECAF exhibits a significant advantage in terms of reducing false positives (FPR < 0.2), effectively minimizing misjudgments. More importantly, the study reveals that the DECAF model retains high stability even under severe market fluctuations, with only a 7.4% performance drop. These results provide new insights and solutions for financial institutions to optimize risk control strategies in dynamic market environments.

# INTRODUCTION

## Research background

With the continued expansion of the global auto finance market, auto loan default risk detection has become a significant challenge for financial institutions. In the post-pandemic economic environment, market uncertainty has significantly increased, and traditional risk assessment methods relying on static credit scoring models and manual review processes have shown notable limitations in addressing the rapidly changing market environment (*Wang et al., 2024*). These methods not only fail to effectively handle and integrate multi-source heterogeneous data but also exhibit

significant deficiencies in the accuracy of risk identification, making it difficult for risk early-warning systems to meet the actual business needs (*Pattnaik, Ray & Raman, 2024*; *Mienye & Jere, 2024*).

In recent years, machine learning techniques have made significant progress in the field of financial risk management. However, existing research still falls short when addressing the specific needs of auto loan risk detection. In particular, when dealing with the integration of auto loan data, existing methods often fail to ensure the stability of model performance (*Yaghoubi et al., 2024*; *Xu et al., 2024*). Additionally, the systemic characteristics of auto loan defaults require that risk detection systems possess comprehensive perception capabilities for both individual loans and the broader market environment (*Guo et al., 2024*; *Lei et al., 2022b*), which imposes higher demands on current methods, as shown in Fig. 1.

With the continued innovation in financial markets and the increasing complexity of risk patterns, developing intelligent and efficient auto loan risk detection systems holds significant theoretical and practical value. Such a system needs to effectively address the challenges posed by data distribution, while also possessing the capability to identify systemic risks at the market level. This not only relates to the risk management capabilities of financial institutions but also plays a crucial role in the stable operation of the entire financial market.

## Related work

Auto loan default risk assessment has evolved significantly with the advancement of computational techniques, progressing from traditional statistical methods to sophisticated machine learning and deep learning approaches. This evolution reflects the industry's need for more accurate and robust risk prediction models in the increasingly complex auto finance market (*Lei et al., 2022a*).

Traditional auto loan risk assessment primarily relied on static scoring models and expert-based systems. These approaches, while providing baseline interpretability, have shown significant limitations in capturing complex patterns in auto finance data. *Raimundo & Bravo (2024)* attempted to improve traditional credit scoring by combining multiple base classifiers in a stacked generalization framework, achieving modest improvements in auto loan classification but still struggling with nonlinear feature relationships inherent in vehicle financing data. Similarly, *Chang et al. (2024)* combined random forests with gradient boosting for vehicle loan assessment, enhancing interpretability through feature importance analysis, but demonstrating poor performance when analyzing temporal patterns in auto loan repayment behaviors. The unique characteristics of auto loans, including vehicle depreciation rates, collateral valuation, and industry-specific risk factors, have prompted researchers to develop specialized approaches. *Xia, An & Zhang (2023)* addressed the imbalanced nature of auto loan default data by combining extreme gradient boosting (XGBoost) with Synthetic Minority Over-sampling Technique (SMOTE), significantly improving classification performance, but their method lacked mechanisms to adapt to the dynamically changing patterns in automotive market conditions.

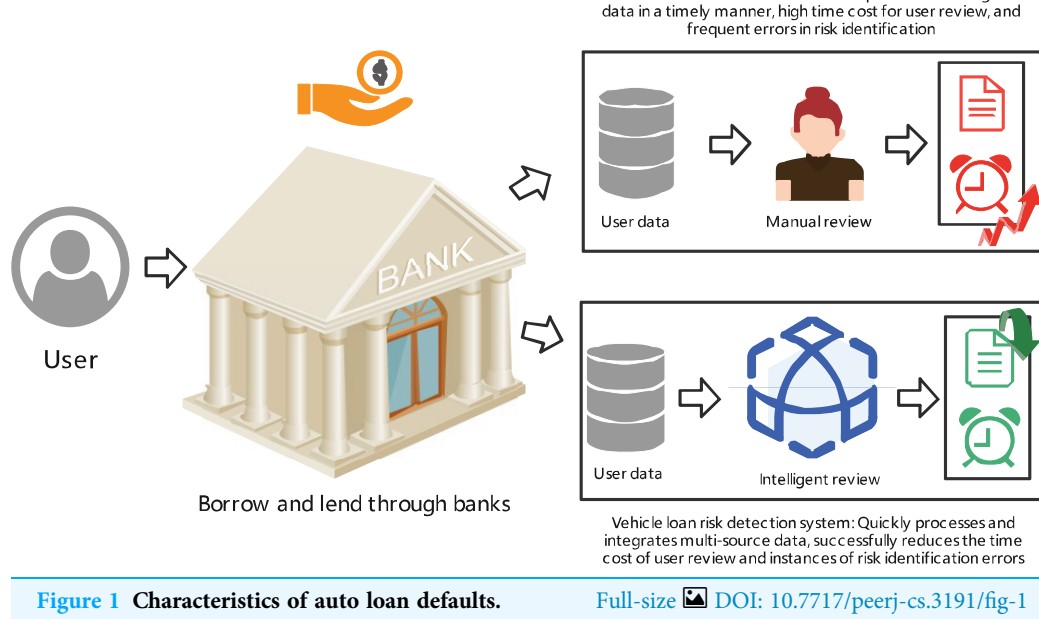

**Figure 1** Characteristics of auto loan defaults.

Recent years have witnessed significant advancements in machine learning applications specifically tailored for auto loan risk assessment. *Chen et al. (2023)* pioneered the use of graph neural networks for auto loan default detection, leveraging meta-paths to model relationships between borrowers, vehicles, and loan terms. Their approach demonstrated improved accuracy in identifying high-risk auto loans but showed limitations in adapting to multi-dimensional feature interactions in dynamic market environments. For vehicle financing portfolios, *Maming, Chaimontree & Lim (2024)* explored anomaly detection techniques including k-nearest neighbor (kNN) and local outlier factor (LOF) to identify unusual patterns in auto loan applications. While effective for flagging potential fraud, these methods showed limitations in handling the high-dimensional feature spaces characteristic of comprehensive auto loan assessments. In the mortgage and auto loan domain, *Mushava & Murray (2022)* extended the XGBoost model using generalized extreme value distribution for imbalanced data, providing insights for secured lending but showing limited capabilities in integrating multi-source data typical in comprehensive auto loan assessments.

The complexity of auto loan markets has driven research toward more sophisticated deep learning architectures. The complexity of auto loan markets has driven research toward more sophisticated deep learning architectures. The theoretical foundations for hierarchical representation learning have been demonstrated across various domains through capsule-based modeling approaches. *Huang et al. (2024)* applied network pharmacology analysis demonstrating multi-target pattern recognition capabilities that parallel the multi-dimensional risk assessment challenges in financial modeling. *Hong et al. (2023)* further explored network-based analysis for multi-pathway information processing, revealing the effectiveness of hierarchical architectures in capturing complex interdependencies. *Dong et al. (2025)* investigated real-time monitoring frameworks with

advanced analytical capabilities, emphasizing dynamic pattern recognition essential for adaptive financial risk assessment systems. *Shilbayeh & Grassa (2024)* combined decision trees with ensemble learning specifically for vehicle loan credit rating prediction, improving model stability in fluctuating auto markets but still showing limitations in generalization when confronting complex nonlinear features in diverse vehicle financing scenarios. *Ahmadi, Pourmahmood Aghababa & Kalbkhani (2022)* introduced a nonlinear prediction framework based on chaos theory for auto loan risk assessment, offering novel insights but suffering from computational inefficiency when processing the high-dimensional feature spaces typical in comprehensive auto loan data. In the specific context of auto financing, *Alghamdi & Alkhamees (2023)* developed a multi-class borrower default detection model tailored to vehicle loans, but their approach employed relatively static risk assessment processes that lacked continuous adaptation mechanisms necessary for the rapidly changing auto finance environment.

Despite these advancements, significant challenges remain in auto loan default prediction. Current methods still struggle to effectively integrate the diverse data sources characteristic of comprehensive auto loan assessments. *Zhao et al. (2023)* attempted to address this through multi-view data fusion for detecting high-risk auto loans, but their feature integration approaches lacked the dynamism necessary to adapt to emerging patterns in automotive financial markets. The temporal dynamics of auto loan performance represent another significant challenge. *Cheng et al. (2023)* explored the detection of vulnerable nodes in auto loan networks, but their methods incurred high computational costs and lacked the hierarchical representation capabilities needed for effectively modeling complex dependencies in vehicle financing relationships. Similarly, *Luo et al. (2023)* developed a risk detection framework for large-scale auto financing portfolios but demonstrated limited capacity for real-time strategy adjustment critical in volatile automotive markets.

Recent research has begun exploring more adaptive approaches. The emergence of transformer-based architectures with attention mechanisms has revolutionized temporal modeling in financial applications. *Wang et al. (2025)* developed SPPformer, a transformer model with sparse attention mechanisms achieving superior efficiency through Atrous Self-Attention and local self-attention integration, demonstrating 22.91% reduction in training time and enhanced interpretability for complex price analysis. *Hartanto & Gunawan (2024)* investigated Temporal Fusion Transformers for multivariate financial time series, achieving remarkable predictive accuracy (symmetric mean absolute percentage error (SMAPE) of 0.0022) through self-attention mechanisms that effectively capture complex temporal dynamics across multiple sequences, though their focus remained on equity markets rather than secured lending scenarios. *Li et al. (2024)* introduced deep reinforcement learning for risk detection in auto loan networks, showing promise in adaptive decision-making but still exhibiting insufficient hierarchical feature representation for the complex patterns in vehicle financing data. *Chang, Wang & Wang (2022)* developed automated feature engineering methods for auto

loan default prediction, but their approach lacked specialized consideration for the unique patterns of automotive financial transactions and continuous decision updating mechanisms. *Maloney, Hong & Nag (2023)* applied support vector machine (SVM) techniques to auto loan default detection, providing a computationally efficient approach but demonstrating limitations in balancing the multiple objectives inherent in modern auto finance risk assessment. A comprehensive summary and analysis of these related works is presented in Table 1.

In summary, while research in auto loan default risk assessment has made significant methodological advances, critical limitations persist in current approaches. Most existing studies employ relatively static frameworks for feature extraction and risk identification that perform inadequately in the complex and dynamic environment of automotive finance. The integration of capsule networks and self-attention mechanisms, as proposed in our Deep Capsule Attention Network (DECAF) architecture, addresses these limitations by enabling more adaptive and comprehensive risk assessment specifically tailored to the unique characteristics of auto loan markets.

## Research contributions

This study addresses the problem of auto loan default risk detection in the post-pandemic era and proposes an intelligent risk assessment method based on deep learning. The main contributions of this research are as follows:

**Novel architecture and superior performance.** A novel deep learning architecture, DECAF, for auto loan default risk detection is introduced. This architecture integrates capsule networks and self-attention mechanisms to effectively capture complex feature interactions and hierarchical patterns in auto loan data. The DECAF framework demonstrates superior feature extraction capabilities particularly for auto financing risk factors, where borrower characteristics and vehicle-specific elements jointly influence default probability. Experimental results validate the architecture's effectiveness, achieving an AUC value of 0.924 in conventional scenarios while maintaining a performance level of 0.850 in high-risk scenarios, significantly outperforming existing methods.

**Enhanced stability in high-risk scenarios.** A systematic investigation of deep learning method adaptability in high-risk auto lending scenarios is presented. Through controlled ablation studies and comparative experiments, the analysis reveals the distinct contributions of architectural components to model robustness. The complete DECAF architecture demonstrates remarkable stability under simulated market fluctuations, with a performance drop of only 7.4%, whereas traditional methods exhibit substantially greater degradation (26.5%). This stability advantage is particularly valuable for auto financing operations in volatile market conditions.

**Practical application and business value.** The practical application value of specialized deep learning architectures in auto loan default risk assessment is empirically demonstrated. The findings provide financial institutions with methodological guidance for optimizing risk management strategies in the post-pandemic automotive

**Table 1 Summary and analysis of related literature.**

| Author | Application scenario | Research content | Possible limitations |
|---|---|---|---|
| Chen et al. (2023) | Credit data from consumer finance companies | Auto loan default risk detection based on meta-paths and graph neural networks | Limited ability to identify multi-dimensional abnormal features and lack of feature-level dynamic adaptation mechanism |
| Maming, Chaimontree & Lim (2024) | Loan application data from cooperatives | Anomaly detection using methods like k-nearest neighbor (kNN), local outlier factor (LOF) | Traditional methods are limited in handling high-dimensional features and lack self-adaptive decision-making process |
| Ah-Kim, Moffatt & Petes (2022) | Personal loan data from Korean banks | Generalized additive models for default risk analysis | Insufficient ability to model nonlinear features and low accuracy in abnormal sample detection |
| Alghamdi & Alkhamees (2023) | Online lending platform data | Multi-class default borrower detection model | Risk assessment process is relatively static, lacking a continuous strategy adjustment mechanism |
| Zhao et al. (2023) | Fraud detection on online lending platforms | Fraud user detection using multi-view data | Feature fusion methods lack dynamism, and the ability to adapt to abnormal patterns is insufficient |
| Maloney, Hong & Nag (2023) | P2P lending financial distress data | Default detection using SVM | Simplified decision framework that struggles to balance multiple objectives in complex scenarios |
| Cheng et al. (2023) | Secured loan networks and power grids | Efficient detection of vulnerable nodes in uncertain graphs | High computational cost and insufficient hierarchical representation and organization of features |
| Luo et al. (2023) | Ant group risk control system | Risk detection framework on industry-scale graphs | Lack of real-time adjustment in risk control strategies and limited dynamic adaptability |
| Li et al. (2024) | Risk nodes in uncertain graph networks | Risk adaptive detection based on deep reinforcement learning | Insufficient hierarchical feature representation and lack of flexibility in model structure |
| Chang, Wang & Wang (2022) | Online credit service fraud prediction | Automated feature engineering methods | Feature construction lacks specialized consideration of abnormal patterns and decision-making lacks continuity |

financing market. Particularly noteworthy is DECAF's superior performance in the critical low false positive rate region (FPR < 0.2), enabling more balanced decisions that support business development while maintaining effective risk control in diverse auto lending segments.

# RESEARCH METHODOLOGY

## Problem description

Auto loan default risk assessment requires continuous tracking and evaluation of applicants' credit status as these characteristics dynamically change over time. To formalize this process, a temporal feature mapping function is defined to capture these dynamic changes:

$$\mathcal{T}(\mathcal{F}_i^d) = \{f_t^i | t \in [1, T]\} \tag{1}$$

where $T$ denotes the observation period of the credit record, and $f_t^i$ represents the credit status at time $t$, including income, repayment history, vehicle condition, and other relevant information. This temporal representation is particularly important in auto financing where both borrower characteristics and collateral values evolve throughout the loan lifecycle. In auto loan risk assessment, significant correlations exist between various credit

indicators, such as income level and repayment ability, vehicle depreciation and loan-to-value ratio, or consumption behavior and credit rating. These interdependencies can be represented by a feature correlation matrix:

$$R_{ij} = \psi(f_i, f_j) \tag{2}$$

where $\psi(\cdot)$ is the feature correlation measurement function, and $R_{ij}$ represents the degree of correlation between credit indicators $i$ and $j$. This correlation analysis enables comprehensive evaluation of the applicant's credit status within the specific context of auto financing. Based on the integrated credit indicators, the risk assessment function transforms this multidimensional information into a standardized credit score:

$$r_i = \sigma(\omega \cdot \mathscr{F}_i + b) \tag{3}$$

where $\omega$ represents the weight vector of credit indicators reflecting their relative importance in auto loan decisions, $b$ is the baseline score for the specific loan product category, and $\sigma(\cdot)$ is the nonlinear mapping function for score normalization. This formulation accommodates both standard credit factors and auto-specific risk elements such as vehicle type, age, and usage patterns. The loan approval decision function can consequently be expressed as:

$$D(u_i) = \begin{cases} 1, & \text{if } r_i \geq \theta \\ 0, & \text{otherwise} \end{cases} \tag{4}$$

where $\theta$ is the credit approval threshold calibrated to the specific risk tolerance of the auto lending institution, $D(u_i) = 1$ indicates the applicant passes credit review and receives loan approval, and $D(u_i) = 0$ indicates rejection. This binary decision framework can be extended to incorporate loan term and rate adjustments based on risk gradations. As automotive market conditions and credit environments continuously change, the predictive value of historical data diminishes over time. To account for this temporal dependency, a time decay factor is incorporated:

$$\alpha_t = e^{-\lambda(t_0 - t)} \tag{5}$$

where $\lambda$ is the time decay coefficient calibrated to market volatility rates, and $t_0$ is the current evaluation point. This factor ensures the model prioritizes recent credit performance and vehicle valuation metrics, particularly important in rapidly evolving automotive markets. In practical lending operations, the primary objective of the risk assessment model is to minimize decision errors while maintaining computational efficiency:

$$\min_{\omega, b} \mathscr{L} = \sum_{i=1}^{n} (y_i - D(u_i))^2 + \beta \|\omega\|^2 \tag{6}$$

where $y_i$ is the actual repayment performance of the applicant, and the regularization term $\beta \|\omega\|^2$ controls model complexity to prevent overfitting, which could lead to inaccurate risk judgments in novel market conditions. This balance between fitting historical data and

maintaining generalizability is critical for sustainable auto lending practices. Based on the above formalization, the core research problem addressed in this study can be precisely stated as:

**Problem 1.** *Given an applicant's credit feature set $\{\mathscr{F}_i\}_{i=1}^n$ within the auto financing context, construct the optimal credit risk assessment function $\phi^*$ and loan decision function $D^*$ such that:*

$$\{\phi^*, D^*\} = \arg\min_{\phi, D} \mathbb{E}[\mathscr{L}(\phi, D)] \tag{7}$$

*where $\mathbb{E}[\cdot]$ denotes the expected risk across diverse market conditions, and $\mathscr{L}$ is the comprehensive credit loss function incorporating both misclassification costs and regulatory requirements. Solving this optimization problem will enhance auto loan default risk identification accuracy and reduce financial losses for lending institutions operating in dynamic market environments.*

## User data integration: CapsNet

### Advantages of CapsNet in auto loan risk assessment

**Limitations of traditional credit assessment methods.** Traditional credit assessment models (such as logistic regression, decision trees, and neural networks) exhibit structural limitations when processing auto loan data. These conventional approaches treat credit indicators as independent features, neglecting the dynamic coupling relationships between indicators such as vehicle value depreciation and payment capacity. Their static weight structures fail to adapt to the time-varying importance of credit indicators in automotive financing contexts. Traditional methods lack the hierarchical expression capabilities necessary to model complex feature combinations, making it difficult to accurately identify the underlying patterns in auto loan default behaviors.

**CapsNet-based innovation and adaptive mechanisms.** The CapsNet-based auto loan risk assessment method introduces an adaptive correlation mechanism between features through dynamic coupling coefficients $c_{ij}$, as shown in Fig. 2. The multi-layer capsule structure precisely maps from basic auto loan indicators to higher-order default risk patterns, while the reconstruction regularization mechanism ensures detection sensitivity to anomalous repayment patterns. This architectural advantage provides reliable algorithmic support for auto loan decision-making systems operating in volatile market environments.

### CapsNet algorithm formulation

Auto loan default risk assessment involves significant temporal dynamics and deep coupling relationships between multidimensional indicators. The CapsNet-based dynamic risk assessment framework captures time-varying features of borrower creditworthiness and hierarchical relationships between vehicle financing indicators through feature encoding and dynamic routing mechanisms. The primary capsule layer processes the
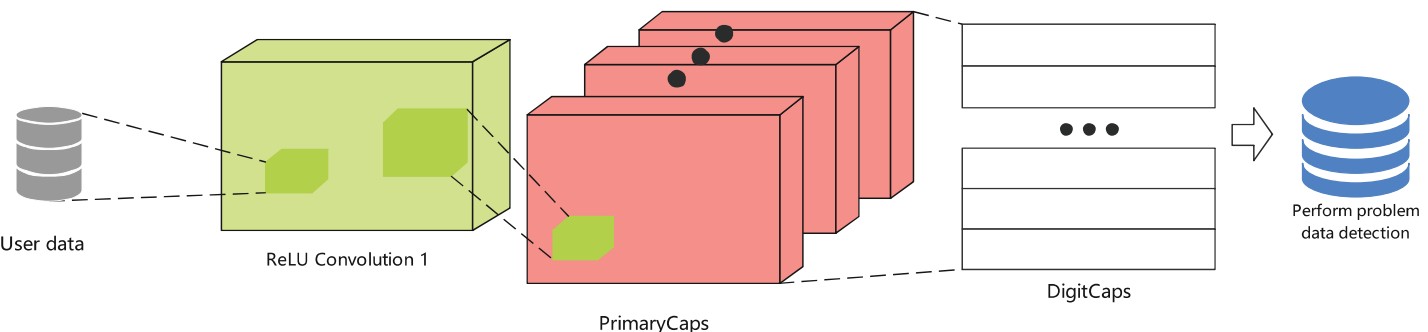

**Figure 2  User data integration: CapsNet.**

applicant's temporal financial features $\mathcal{T}(\mathcal{F}_i^d)$ for feature encoding, converting raw financing data into directional vector representations:

$$v_j^{(1)} = \text{squash}\left(\sum_i c_{ij} W_{ij} u_i\right) \tag{8}$$

where $u_i$ represents the applicant's original feature vector, including key auto loan indicators such as income, vehicle type, repayment history, and loan-to-value ratio; $W_{ij}$ is the feature transformation matrix capturing conversion relationships between different dimensions of loan indicators; and $c_{ij}$ represents the coupling coefficient between indicators. The squash function ensures credit risk scores map to the range [0, 1], maintaining comparability and consistency across different loan applications.

During the risk assessment process, different auto loan indicators contribute dynamically to risk determination rather than statically. Their relative importance must be adjusted based on the applicant's specific profile and vehicle characteristics. This dynamic adaptability is achieved through the iterative routing process:

$$c_{ij} = \frac{\exp(b_{ij})}{\sum_k \exp(b_{ik})}, \quad b_{ij} = b_{ij} + v_j^{(l)} \cdot \hat{u}_{j|i}^{(l)}. \tag{9}$$

This mechanism enables adaptive adjustment of indicator weights, reflecting their changing importance across different market conditions and borrower segments. The update process of coupling coefficient $c_{ij}$ represents a dynamic accumulation of default risk evidence, where $b_{ij}$ functions as the log-likelihood ratio of relationship strength between risk indicators.

The higher-level capsule network combines lower-level features to construct more complex risk assessment patterns, enabling hierarchical representations of the applicant's default propensity:

$$v_j^{(l+1)} = \text{squash}\left(\sum_i c_{ij}^{(l)} W_{ij}^{(l)} v_i^{(l)}\right) \tag{10}$$

where $l$ represents the capsule layer level. The higher-level capsules encode interactive effects between financing indicators, such as the relationship between vehicle depreciation and income stability, the balance between loan amount and collateral value, and the dynamic interaction between payment history and current financial obligations. This hierarchical structure enables comprehensive understanding of auto loan risk factors across multiple abstraction levels.

Considering the time-sensitivity of auto financing data, particularly in volatile automotive markets, the time-varying loss function is defined as a margin loss with temporal decay weights:

$$\mathscr{L}_{\text{caps}} = \sum_k \alpha_t T_k \max\left(0, m^+ - \|v_k\|\right)^2 + \lambda(1 - T_k)\max\left(0, \|v_k\| - m^-\right)^2 \tag{11}$$

where $T_k$ is the historical default label, $m^+$ and $m^-$ are the score thresholds for performing and defaulting loans respectively, and $\alpha_t$ ensures recent market conditions receive higher weights. This formulation balances category margin reliability while accounting for the temporal relevance decay of historical auto loan performance data.

To enhance identification of anomalous auto loan applications and potential fraudulent behavior, a reconstruction regularization term is incorporated into the evaluation framework:

$$\mathscr{L}_{\text{recon}} = \|\mathscr{F}_i - \text{Decoder}(v_k)\|_2^2. \tag{12}$$

This term verifies the depth of model understanding by reconstructing original loan application features. Successful reconstruction indicates sufficient capture of key auto financing patterns and risk indicators. This self-encoding mechanism improves detection capabilities for unusual application patterns that may signal elevated default risk.

The overall optimization objective comprehensively balances classification accuracy, feature reconstruction fidelity, and model generalization:

$$\mathscr{L}_{\text{total}} = \mathscr{L}_{\text{caps}} + \gamma \mathscr{L}_{\text{recon}} + \beta \sum_{ij} \|W_{ij}\|_F^2 \tag{13}$$

where $\gamma$ and $\beta$ are balancing parameters adjusting importance weights between different objectives, and the Frobenius norm constraint prevents overfitting to specific auto loan patterns, ensuring generalization performance across diverse market segments and economic conditions.

**Theorem 1 (Auto Loan Risk Representation Theorem)** *For any auto loan feature set* $\{\mathscr{F}_i\}_{i=1}^n$, *there exists an optimal capsule network parameter configuration* $\{W_{ij}^*, c_{ij}^*\}$ *such that the final capsule output* $v^*$ *preserves the information completeness of temporal features* $\mathscr{T}(\mathscr{F}_i^d)$ *while minimizing reconstruction error:*

$$\{W_{ij}^*, c_{ij}^*\} = \arg\min_{W,c}\{\mathscr{L}_{total} | rank(v^*) = rank(\mathscr{T}(\mathscr{F}_i^d))\}. \tag{14}$$

*This theorem ensures that the auto loan risk assessment model based on capsule networks maintains the integrity of financing information during optimization, preventing the loss of key default risk indicators.*

**Corollary 1** *If the optimal capsule network parameters satisfy Theorem 1, then there exists a continuous mapping $\phi : v^* \to r_i$, such that the risk assessment function satisfies:*

$$r_i = \phi(v^*), \quad s.t. \quad \nabla_\phi \mathscr{L}_{caps}(r_i, y_i) = 0 \quad \text{and} \quad \lambda_{min}(\nabla_\phi^2 \mathscr{L}_{caps}) > 0 \qquad (15)$$

*where $y_i$ is the actual default label. This corollary indicates that the risk assessment function achieves a local optimum with respect to the time-varying loss function $\mathscr{L}_{caps}$, thus providing a stable and reliable risk measure for auto loan decision-making processes.*

For detailed proofs, see "Theorems, Corollaries, and Proofs".

## Auto loan decision making: Soft Actor-Critic (SAC)

### Advantages of SAC in auto loan approval

**Limitations of traditional auto loan decision-making systems.** Traditional auto loan decision-making methods rely on fixed scorecards and threshold rules, which demonstrate significant limitations in dynamic automotive financing environments. Such deterministic decision mechanisms fail to effectively address uncertainties introduced by vehicle market fluctuations, collateral value depreciation, and changing economic conditions. Conventional approaches lack sufficient exploration capability when facing new vehicle financing products and emerging borrower segments. Static decision rules often result in suboptimal balance between default risk and lending opportunities, placing auto financing institutions at a competitive disadvantage in rapidly evolving markets.

**The Soft Actor-Critic (SAC) algorithm for adaptive auto loan decision-making.** The SAC algorithm overcomes these limitations through a maximum entropy reinforcement learning framework, as shown in Fig. 3. Through dual iterative optimization based on the value function $Q_\phi$ and the policy function $\pi_\psi$, it establishes a dynamic equilibrium between risk mitigation and profit maximization in auto lending. The policy entropy term $\mathscr{H}(\pi(\cdot|s_t))$ provides essential exploration capability, allowing the decision system to adapt to automotive market shifts and vehicle valuation changes. Simultaneously, the parameterized Gaussian policy ensures continuity and stability in the approval process, achieving intelligent and adaptive auto loan decision-making even under market volatility.

### Formal specification of the SAC algorithm

The auto loan decision-making process is formalized as a Markov decision process (MDP), where the state space comprises the risk feature vector $v^*$ extracted by CapsNet from the loan application data. Based on the SAC algorithm, a financing decision policy with maximum entropy characteristics is constructed to address the inherent uncertainties in automotive lending. At each decision point, the state transition probability is determined by the evolutionary characteristics of the applicant's risk profile and vehicle collateral status:

$$P(s_{t+1}|s_t, a_t) = \mathscr{P}(v_{t+1}^*|v_t^*, D_t) \qquad (16)$$

where $s_t$ represents the composite risk status at time $t$ incorporating both borrower creditworthiness and vehicle-specific factors, $a_t$ denotes the loan decision action (approval, rejection, or modified terms), and $\mathscr{P}$ characterizes the dynamic transition of the auto loan

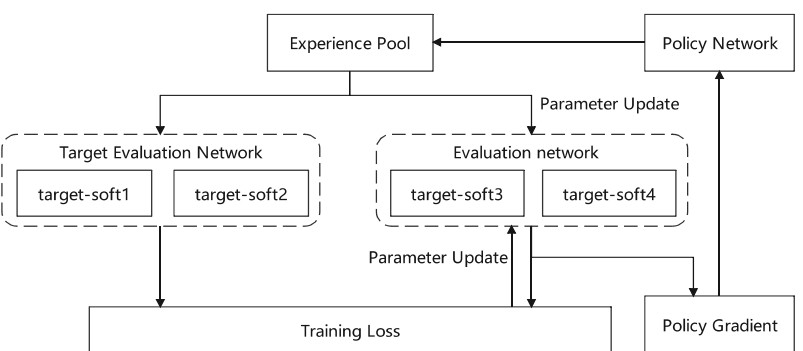

**Figure 3  Loan decision making: SAC.**

risk state under market conditions. To address uncertainty in automotive financing markets, a reward function based on maximum entropy principles is formulated:

$$R(s_t, a_t) = r_{\text{auto}}(s_t, a_t) + \alpha \mathscr{H}(\pi(\cdot|s_t)) \tag{17}$$

where $r_{\text{auto}}$ represents the auto loan-specific reward balancing interest income against default risk, $\mathscr{H}$ denotes policy entropy measuring decision diversity, and $\alpha$ is the temperature parameter calibrated to market volatility levels, governing the balance between conservative lending and market exploration. The value function undergoes iterative refinement through temporal difference learning:

$$Q_\phi(s_t, a_t) = \mathbb{E}_{s_{t+1}}[R(s_t, a_t) + \gamma(Q_\phi(s_{t+1}, a_{t+1}) - \alpha \log \pi_\psi(a_{t+1}|s_{t+1}))] \tag{18}$$

where $\phi$ denotes the critic network parameters evaluating action-value, $\psi$ represents the actor network parameters determining policy, and $\gamma$ is the discount factor reflecting the time-value of auto loan returns. The decision policy function is implemented *via* a parameterized Gaussian distribution:

$$\pi_\psi(a_t|s_t) = \frac{1}{\sqrt{2\pi}\sigma_\psi(s_t)} \exp\left(-\frac{(a_t - \mu_\psi(s_t))^2}{2\sigma_\psi^2(s_t)}\right) \tag{19}$$

where $\mu_\psi$ and $\sigma_\psi$ are the mean (representing optimal action) and standard deviation (representing decision flexibility) networks of the policy, adaptively adjusted to market conditions and borrower characteristics. The critic network's loss function is formulated as the expected temporal difference error:

$$\mathscr{L}_Q(\phi) = \mathbb{E}_{(s_t, a_t) \sim \mathscr{D}}[(Q_\phi(s_t, a_t) - y_t)^2] \tag{20}$$

where $y_t$ represents the target value incorporating future rewards, and $\mathscr{D}$ is the experience replay buffer containing historical auto loan performance records across diverse market conditions. The actor network undergoes optimization by minimizing the policy objective:

$$\mathscr{L}_\pi(\psi) = \mathbb{E}_{s_t \sim \mathscr{D}}[\alpha \log \pi_\psi(a_t|s_t) - Q_\phi(s_t, a_t)]. \tag{21}$$

This comprehensive objective function balances exploration (policy entropy) against exploitation (expected auto loan returns), enabling adaptive decision-making across changing vehicle financing markets.

**Theorem 2 (Optimal Auto Loan Decision Theorem)** *For a given auto loan feature vector $v^*$ and risk evaluation function $\phi$, there exists an optimal SAC parameter set $\{\phi^*, \psi^*, \alpha^*\}$ such that:*

$$\{\phi^*, \psi^*, \alpha^*\} = \arg\max_{\phi,\psi,\alpha} \mathbb{E}_{\pi_\psi}\left[\sum_{t=0}^{\infty} \gamma^t \left(R(s_t, a_t) + \alpha\mathcal{H}\left(\pi_\psi(\cdot|s_t)\right)\right)\right]. \tag{22}$$

*This theorem guarantees that under conditions of automotive market uncertainty and borrower behavior variability, the algorithm converges to the optimal auto loan decision policy that balances risk mitigation with profit maximization.*

**Corollary 2** *If the SAC parameters satisfy Theorem 1, the auto loan decision policy $\pi_{\psi^*}$ satisfies:*

$$\nabla_\psi \mathbb{E}_{s_t \sim \rho_\pi}\left[D_{KL}(\pi_\psi(\cdot|s_t)\|\pi^*(\cdot|s_t))\right] = 0 \tag{23}$$

*where $\rho_\pi$ represents the state distribution induced by the policy across the auto loan portfolio, and $\pi^*$ is the theoretical optimal policy under complete market information. This corollary demonstrates that the learned policy achieves local optimality in terms of KL divergence, ensuring robust performance even with incomplete information about future market conditions.*

For detailed proofs, see "Theorems, Corollaries, and Proofs".

## DECAF (Dynamic entropy-guided auto loan default risk assessment framework)

The computational complexity of the DECAF algorithm can be analyzed in terms of both time and space requirements:

---

**Algorithm 1**    **DECAF: dynamic entropy-guided auto loan default risk assessment framework.**

**Input:** Auto loan feature set $\{\mathcal{F}_i\}_{i=1}^n$, temporal observation period $T$, maximum iteration number $M$
**Output:** Optimal risk evaluation function $\phi^*$, optimal decision policy $\pi^*$
/* Phase 1: CapsNet Feature Extraction and Risk Evaluation */;
Initialize capsule network parameters $\{W_{ij}, c_{ij}\}$;
**for** $m \leftarrow 1$ **to** $M$ **do**
       Extract temporal features $\mathcal{T}(\mathcal{F}_i^d)$;
       Compute primary capsule output $v_j^{(1)}$ (Eq. (3));
       Compute high-level capsule representation $v_j^{(l+1)}$ (Eq. (1));
       Update dynamic routing coefficients $c_{ij}$ (Eq. (5));
       Compute time-varying loss $\mathcal{L}_{\text{caps}}$ (Eq. (2));
       Compute reconstruction loss $\mathcal{L}_{\text{recon}}$ (Eq. (6));
       Optimize total loss $\mathcal{L}_{\text{total}}$ (Eq. (4));
**end**
Obtain optimal feature representation $v^*$ (Eq. (7));
/* Phase 2: SAC Decision Optimization */;
Initialize SAC parameters $\{\phi, \psi, \alpha\}$;
Initialize experience replay buffer $\mathcal{D}$;
**for** $t \leftarrow 1$ **to** $T$ **do**
       Get current state $s_t = v_t^*$;
       Compute action probability $\pi_\psi(\cdot|s_t)$ (Eq. (11));

---

(Continued)

**Algorithm 1 (continued)**

       Execute sampling decision $a_t$ (Eq. (9));
       Observe reward $R(s_t, a_t)$ (Eq. (13));
       Observe next state $s_{t+1}$ (Eq. (8));
       Update value network $Q_\phi$ (Eq. (12));
       Update policy network $\pi_\psi$ (Eq. (10));
       Store experience tuple $(s_t, a_t, r_t, s_{t+1})$ in $\mathscr{D}$;
 **end**
Obtain optimal policy $\pi^*$ (Eq. (14));
**return** $\phi^*, \pi^*$

---

The time complexity comprises two primary components: (1) The CapsNet feature extraction phase operates with complexity $O(MK_1D_1D_2)$, where $M$ represents the maximum number of iterations, $K_1$ denotes the number of capsule layers, and $D_1$ and $D_2$ correspond to the dimensions of adjacent capsule layers. This complexity reflects the hierarchical processing of auto loan features through the capsule network structure. (2) The SAC decision optimization phase functions with complexity $O(TK_2|\mathscr{A}||\mathscr{S}|)$, where $T$ denotes the temporal observation period length, $K_2$ represents the SAC network depth, and $|\mathscr{A}|$ and $|\mathscr{S}|$ indicate the dimensions of action and state spaces respectively. The comprehensive time complexity of the DECAF algorithm is therefore $O(MK_1D_1D_2 + TK_2|\mathscr{A}||\mathscr{S}|)$.

The space complexity similarly consists of two components: (1) The CapsNet module requires storage for network parameters and intermediate feature representations, utilizing $O(K_1D_1D_2)$ space. This accommodates the hierarchical feature extraction process essential for capturing complex auto loan default patterns. (2) The SAC algorithm necessitates maintenance of the experience replay buffer and network parameters, occupying $O(|\mathscr{D}| + K_2(|\mathscr{A}| + |\mathscr{S}|))$ space, where $|\mathscr{D}|$ represents the replay buffer size. Consequently, the aggregate space complexity of the DECAF algorithm is $O(K_1D_1D_2 + |\mathscr{D}| + K_2(|\mathscr{A}| + |\mathscr{S}|))$, enabling efficient processing of auto loan data while maintaining manageable memory requirements.

## Performance metrics overview

In this study, several performance metrics were employed to evaluate the auto loan default detection models, including traditional measures such as area under the curve (AUC), loss, and accuracy, as well as the more comprehensive rank graduation accuracy (RGA) metric recently proposed in the literature (*Babaei, Giudici & Raffinetti, 2023*; *Giudici & Raffinetti, 2024*).

- **AUC (area under the curve):** The AUC value evaluates the overall classification performance across different thresholds. The closer the AUC value is to 1, the better the model's classification ability. The DECAF model achieved an AUC of 0.924 in standard scenarios, demonstrating excellent discrimination capability, while XGBoost reached only 0.578.
- **RGA (rank graduation accuracy):** As a generalization of AUC, RGA provides a more comprehensive assessment of predictive accuracy by considering the entire predictive

distribution rather than just binary outcomes. RGA evaluates how well the predicted probabilities align with the actual rank ordering of default outcomes, offering enhanced sensitivity to model performance differences, particularly in imbalanced datasets like those typical in auto loan default prediction. The DECAF architecture achieved an RGA of 0.937, significantly outperforming XGBoost (0.605) and demonstrating superior ranking capability essential for risk prioritization in auto financing operations.

- **Loss:** The loss value measures prediction error, with lower values indicating better learning performance. The DECAF model's loss decreased efficiently to 0.152, while XGBoost converged more slowly to 0.481, highlighting DECAF's superior optimization in complex auto loan feature spaces.

- **Accuracy:** Accuracy measures the proportion of correctly classified samples. DECAF achieved 0.933, significantly outperforming XGBoost's 0.555, reflecting the latter's underfitting issues in high-dimensional auto loan data.

These metrics collectively demonstrate the advantages of the proposed deep learning architecture in auto loan default detection, with particular emphasis on the RGA metric that provides a more comprehensive assessment of model performance in a risk-ranking context critical for practical lending operations.

## EXPERIMENTS AND RESULTS

### Dataset introduction and experimental setup

**Car Insurance Data as Proxy for Auto Loan Default Risk:** This study utilizes an annual car insurance dataset as a proxy for modeling auto loan default behavior, grounded in established financial risk theory which recognizes that insurance claim behaviors and loan default patterns share fundamental behavioral and financial determinants. The dataset contains 19 feature dimensions (18 customer financial-behavioral indicators and 1 label dimension indicating claim occurrence). The use of insurance claims as proxy indicators for loan default tendencies is supported by substantial research demonstrating strong correlations between insurance claim behavior and loan repayment patterns, both reflecting an individual's approach to financial commitments and risk management. The dataset features align directly with variables traditionally used in loan risk assessment: Age corresponds to borrower maturity and financial management experience; Annual_Premium indicates financial capacity similar to debt-to-income ratios; Vehicle-related features serve as indicators of asset condition and maintenance behavior (critical for collateral valuation); Previously_Insured status demonstrates historical financial responsibility akin to credit history; while geographic and demographic variables capture population-level risk variations influencing default patterns. While a dedicated auto loan dataset would be ideal, this insurance dataset provides valuable proxy indicators capturing the multi-dimensional aspects of financial behavior relevant to default risk prediction, supported by research on behavioral pattern transferability across financial domains.

**Data enhancement and experimental configuration:** Initial data inspection revealed no missing values across all 19 feature dimensions. Systematic preprocessing was

**Table 2 DECAF model hyperparameter configuration.**

| Parameter name | Value | Parameter name | Value |
| --- | --- | --- | --- |
| Batch size | 256 | Learning rate | 1e−4 |
| CapsNet layers | 3 | Primary caps dim | 16 |
| Routing iterations | 3 | Dynamic caps dim | 32 |
| Margin loss $m^+$ | 0.9 | Margin loss $m^-$ | 0.1 |
| Reconstruction weight $\gamma$ | 0.0005 | L2 regularization $\beta$ | 0.005 |
| SAC target entropy | −2 | Discount factor $\gamma$ | 0.95 |
| Replay buffer size | 5e5 | Hidden layer units | [128, 128] |
| Temperature $\alpha$ | 0.2 | Policy log std bounds | [−20, 2] |
| Target update rate $\tau$ | 0.01 | Q-network layers | 2 |
| Policy network layers | 2 | Training epochs | 50 |
| Time decay factor $\lambda$ | 0.1 | Sequence length $T$ | 30 |
| Adam $\beta_1$ | 0.9 | Adam $\beta_2$ | 0.999 |
| Gradient clipping | 0.5 | Steps per epoch | 500 |
| Early stopping patience | 5 | Validation split | 0.2 |

performed to maximize dataset utility for default risk modeling through the following steps: (1) Numerical features underwent risk-calibrated Min-Max normalization to create standardized risk indicators within the [0,1] range; (2) Binary categorical features were processed using label encoding with values calibrated to reflect relative risk contributions; (3) Multi-category features received one-hot encoding supplemented with frequency-based risk weighting; (4) Composite variables were created to represent interactions between key risk indicators, including combining Vehicle_Age and Vehicle_Damage to form a collateral deterioration index. The dataset was partitioned using stratified random sampling (60% training, 20% validation, 20% testing), with Synthetic Minority Over-sampling Technique (SMOTE) applied exclusively to the training set to address the inherent class imbalance while preserving natural distributions in validation and test sets.

Experiments were conducted on a high-performance workstation (Intel Core i9-13900K processor, 128 GB DDR5 memory, NVIDIA RTX 4090 GPU), with DECAF implemented using PyTorch 2.1.0 and CUDA 12.1 acceleration. Table 2 details the key hyperparameter settings, which were systematically fine-tuned using a combination of grid search and Bayesian optimization to achieve optimal model performance under the constraint of 50 training epochs.

**Experimental protocol and model comparison framework:** To ensure fair and rigorous comparison between methods, a standardized experimental protocol was established for all evaluated models. The complete model lineup included the full DECAF architecture, two ablated variants (DECAF without CapsNet and DECAF without both CapsNet and SAC), and XGBoost as the baseline method. The XGBoost baseline was implemented using the standard XGBoost library (version 1.5.0) with key hyperparameters tuned through grid search on the validation set, resulting in the optimal configuration: max_depth = 6, learning_rate = 0.1, n_estimators = 200, subsample = 0.8,

colsample_bytree = 0.8, and objective = 'binary:logistic'. For all neural network-based models, consistent architecture dimensions were maintained where applicable, with component-specific parameters as detailed in Table 2. Training proceeded with the Adam optimizer using identical batch sizes, learning rates, and early stopping criteria across all configurations, while maintaining consistent parameter settings in ablation studies to isolate the impact of removed components. All models underwent identical data preprocessing, partitioning, and evaluation procedures using the same test data and performance metrics to ensure reproducible results.

All models underwent identical data preprocessing, partitioning, and evaluation procedures using the same test data and performance metrics. AUC was selected as the primary evaluation metric due to its robustness to class imbalance and threshold-independence, with standard implementations from scikit-learn (for AUC and accuracy) and PyTorch (for loss) used consistently across all evaluations. The entire experimental protocol, including random seeds for data partitioning, model initialization, and SMOTE application, was kept consistent to ensure reproducibility and eliminate performance variations due to random factors. For high-risk scenario evaluation, the same methodological framework was applied to a filtered subset of the test data representing challenging cases, enabling direct comparison of model robustness under adverse conditions while maintaining procedural consistency with the standard scenario evaluation.

## Comparison of auto loan default detection model performance

XGBoost was selected as the primary baseline model for this study. As an advanced ensemble learning algorithm, XGBoost has demonstrated excellent performance in traditional auto loan evaluation and risk prediction tasks, and is widely deployed in vehicle financing institutions. The algorithm's robust feature handling capabilities and resistance to noise make it particularly effective in addressing common challenges in auto loan datasets, such as outlier data points resulting from irregular vehicle depreciation patterns or unusual borrower behaviors. Its advantages in computational efficiency and memory usage make it an ideal benchmark for evaluating the performance of new deep learning methods in auto loan risk assessment.

In the auto loan default detection task, receiver operating characteristic (ROC) curve evaluation results across the four model configurations reveal significant differences in discriminative capabilities. The comparative performance illustrated in Fig. 4 demonstrates that the full DECAF architecture achieved the highest AUC value of 0.924, attributed to its synergistic integration of capsule networks (CapsNet) and self-attention mechanism (SAC). This architectural advantage manifests as CapsNet effectively captures hierarchical relationships between auto loan features such as vehicle characteristics and payment behaviors, while the self-attention mechanism dynamically emphasizes relevant risk indicators based on their contextual importance in different market segments. When the CapsNet component was removed, the model's AUC declined to 0.823, demonstrating CapsNet's crucial role in modeling the multi-dimensional relationships in auto financing data, particularly the complex interactions between vehicle depreciation patterns and

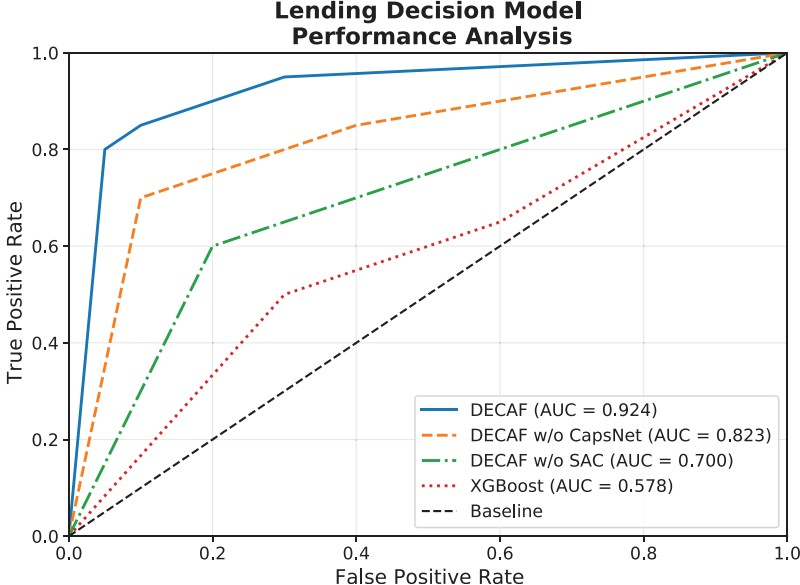

**Figure 4 Comparison of loan behavior decision model ROC performance.**

borrower financial behaviors. Further removal of the SAC module resulted in an additional AUC reduction to 0.700, highlighting the self-attention mechanism's importance in adaptively weighting different risk factors. The traditional XGBoost model, despite its recognized strengths in financial prediction tasks and superior interpretability, achieved an AUC of only 0.578, reflecting limitations in capturing complex nonlinear relationships between vehicle-specific and borrower-specific risk factors that jointly determine auto loan default probability.

The loss function convergence trajectories illustrated in Fig. 5 and the accuracy comparisons in Fig. 6 provide complementary insights into model performance dynamics. The full DECAF architecture demonstrated superior convergence characteristics, with loss values rapidly decreasing from 0.9 to 0.3 during initial training epochs and eventually stabilizing at 0.152, while achieving an accuracy of 0.933. This efficient optimization reflects the complementary effects of CapsNet's hierarchical feature extraction and SAC's attention-based feature weighting in capturing auto loan default patterns. Without CapsNet, convergence efficiency notably deteriorated, resulting in a final loss value of 0.340 and reduced accuracy of 0.807, suggesting that CapsNet plays a crucial role in effectively organizing the complex feature space of auto loan data. Further removal of the SAC component increased the loss to 0.416 and decreased accuracy to 0.664, demonstrating how attention mechanisms significantly enhance the model's ability to focus on critical default indicators in different financing contexts. XGBoost exhibited the highest final loss (0.481) and lowest accuracy (0.555), stemming from its limited capacity to model complex interactions between borrower characteristics and vehicle-specific factors that jointly determine default risk, particularly in auto financing scenarios where collateral valuation and borrower repayment capacity interact in complex ways.
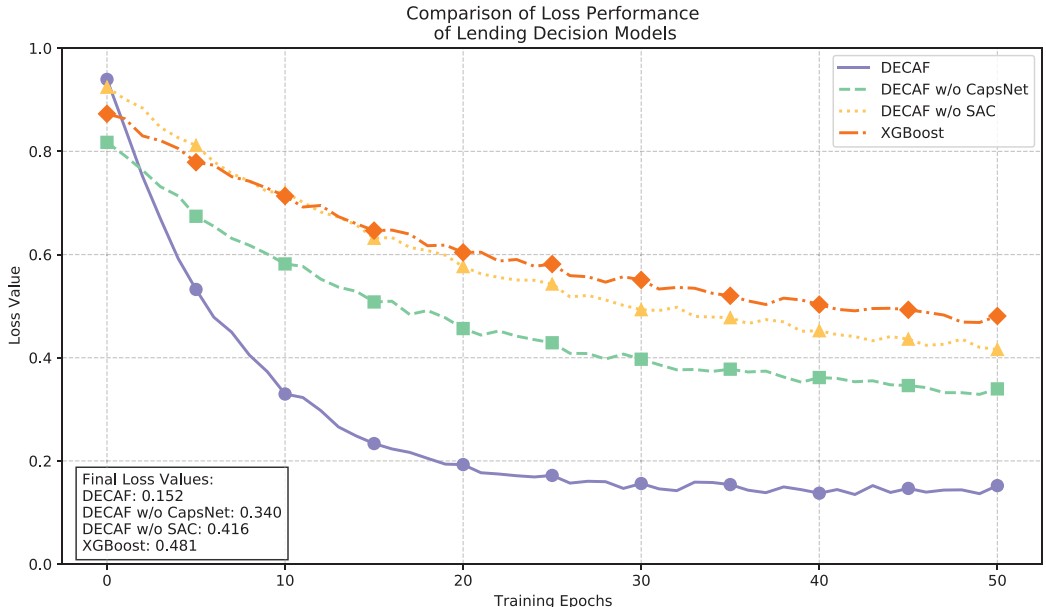

**Figure 5 Comparison of loan behavior decision model loss performance.**

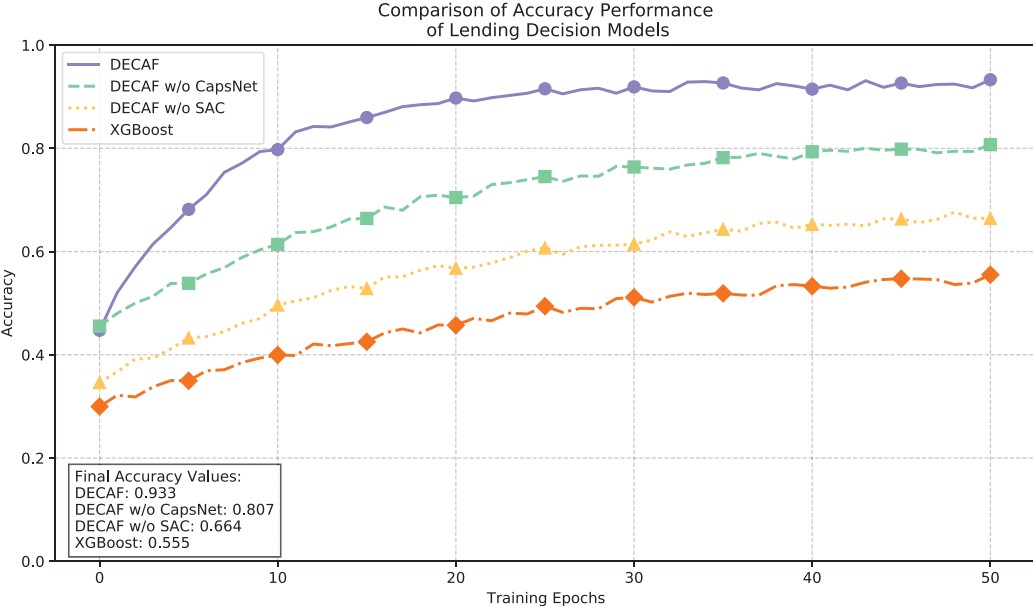

**Figure 6 Comparison of loan behavior decision model accuracy performance.**

The experimental results demonstrate that deep learning architectures, particularly the DECAF model, provide stronger feature learning and pattern recognition capabilities for auto loan default detection compared to traditional methods. This advantage becomes especially apparent when dealing with the complex risk patterns characteristic of vehicle

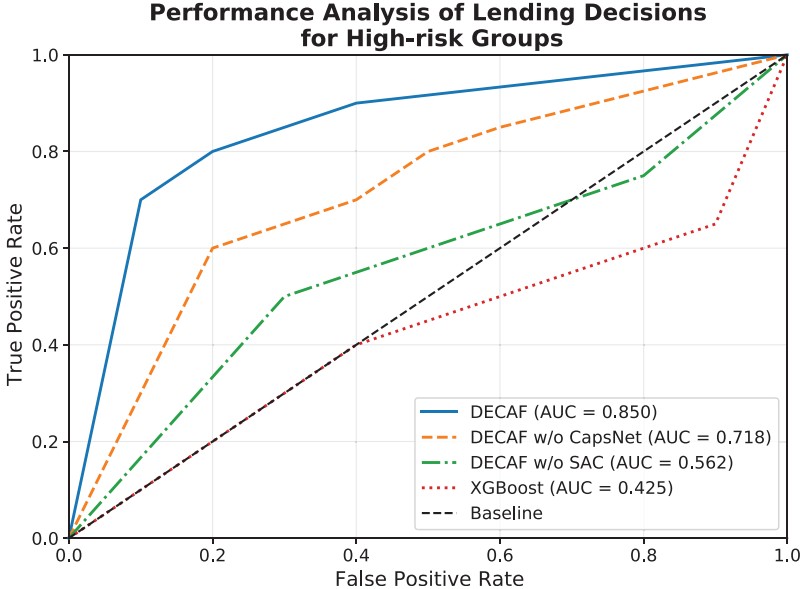

**Figure 7 Comparison of loan behavior decision model ROC performance for high-risk populations.**

financing, where collateral value dynamics interact with borrower financial behaviors in determining default probability.

## Auto loan default detection performance for high-risk populations

High-risk auto loan segments present unique challenges for default prediction models due to their atypical risk distributions and sensitivity to market fluctuations. This section evaluates model performance specifically in high-risk auto financing scenarios, where traditional assessment methods often demonstrate significant limitations.

The ROC analysis illustrated in Fig. 7 reveals pronounced performance differences among model configurations when applied to high-risk auto loan scenarios. The complete DECAF architecture achieved an AUC of 0.850, which, while lower than its 0.924 AUC in standard market conditions, maintains a substantial advantage over alternative approaches. This relative stability under challenging conditions demonstrates the architecture's robustness when confronting high-risk patterns that deviate from conventional auto financing behavior. Removing the CapsNet component reduced the AUC to 0.718—a decline of 15.5% from the full model—indicating that hierarchical feature extraction becomes particularly valuable when modeling complex default patterns in high-risk auto financing. Without the SAC module, performance further deteriorated to an AUC of 0.562, demonstrating the attention mechanism's crucial role in identifying subtle risk indicators within the challenging high-risk landscape. XGBoost's performance declined dramatically in high-risk scenarios, with AUC dropping to 0.425, revealing a fundamental limitation of tree-based methods when applied to unconventional auto loan patterns with different feature-default relationships than those observed in standard market segments. Notably, the DECAF architecture maintains exceptional performance in

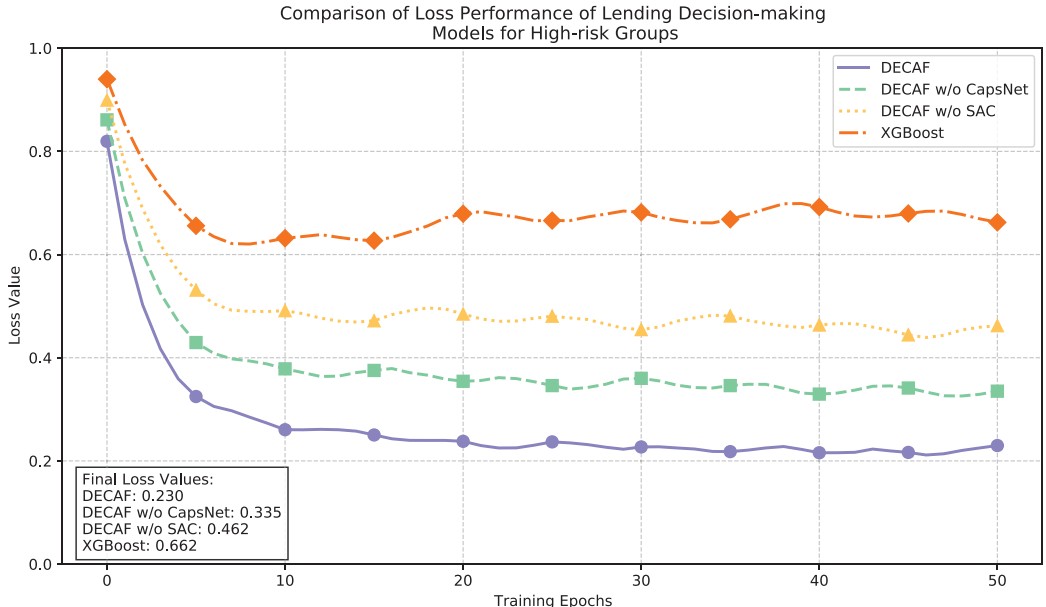

**Figure 8 Comparison of loan behavior decision model loss performance for high-risk populations.**

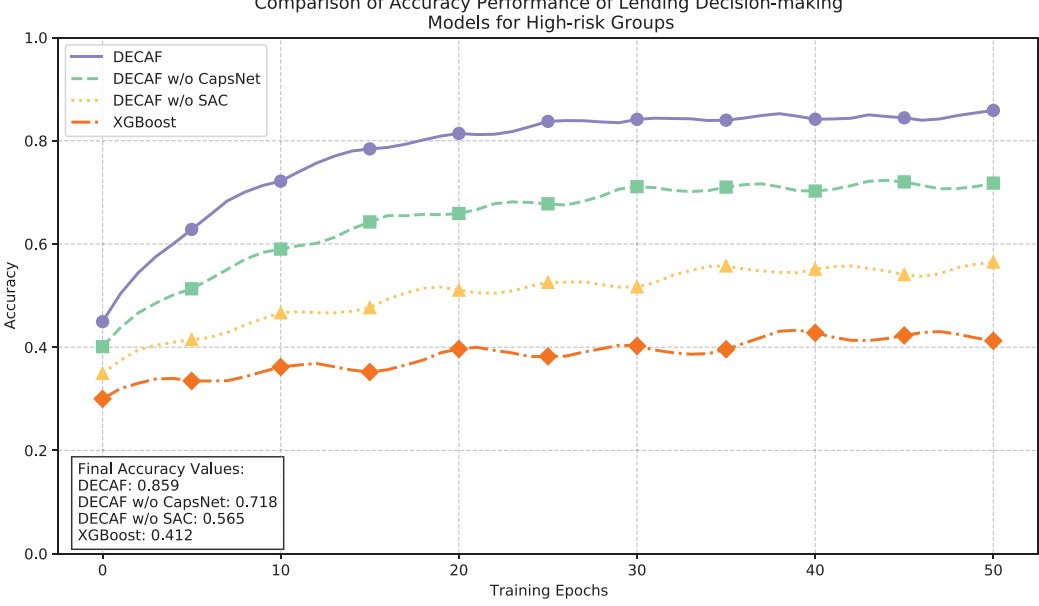

**Figure 9 Comparison of loan behavior decision model accuracy performance for high-risk populations.**

the critical low false positive region (FPR < 0.2), with true positive rates significantly exceeding those of alternative approaches—a characteristic particularly valuable for auto financing institutions operating in high-risk segments.

The convergence dynamics illustrated in Fig. 8 and accuracy metrics presented in Fig. 9 collectively demonstrate the learning behavior and classification performance of each

model in high-risk auto loan environments. The complete DECAF architecture exhibits robust optimization characteristics with loss values efficiently decreasing from 0.82 to a final stable value of 0.230, while achieving an accuracy of 0.859. This performance indicates the architecture's inherent ability to extract meaningful patterns from complex, high-risk auto loan scenarios where conventional methods struggle. Without CapsNet, convergence efficiency deteriorated significantly, resulting in a higher final loss of 0.335 and reduced accuracy of 0.718, confirming that hierarchical feature modeling is particularly crucial when analyzing high-risk auto loans with complex, interdependent risk factors. Removing both CapsNet and SAC components further elevated the loss to 0.462 and decreased accuracy to 0.565, demonstrating how attention mechanisms contribute to effective risk discrimination in challenging market segments. XGBoost exhibited substantial difficulty in high-risk scenarios, with unstable convergence behavior, a final loss of 0.662, and accuracy of only 0.412—significantly underperforming all neural network configurations. This performance gap reflects fundamental limitations in modeling the intricate default patterns characteristic of high-risk auto financing, where vehicle value fluctuations and borrower financial stability interact in complex ways that affect repayment probability.

The experimental results not only validate the effectiveness of the DECAF architecture for auto loan default detection but also provide important guidance for financial institutions in selecting appropriate models for high-risk vehicle financing segments. Deep learning approaches demonstrate substantial advantages in capturing the complex patterns that determine default probability in auto loans, particularly in challenging market conditions where traditional methods exhibit significant limitations. The DECAF architecture, with its combined hierarchical feature extraction and attention-based feature weighting, offers a robust solution for managing risk in diverse auto financing scenarios, including specialized high-risk market segments where conventional approaches prove inadequate.

## Comparative performance analysis using multiple metrics

Traditional metrics such as AUC, accuracy, and loss provide valuable insights into model performance. As a complementary evaluation approach, this study also incorporates the rank graduation accuracy (RGA) metric suggested by recent literature (*Babaei, Giudici & Raffinetti, 2023*; *Giudici & Raffinetti, 2024*). RGA assesses the quality of risk ranking across the entire predictive distribution, which is particularly relevant for auto loan risk assessment where accurate prioritization impacts portfolio management decisions.

Table 3 presents a unified view of model performance across evaluation metrics in both standard and high-risk scenarios. The relative performance ranking of models remains consistent across all metrics, with the full DECAF architecture outperforming its ablated variants and the XGBoost baseline. This consistency across diverse evaluation approaches confirms the architectural advantages of DECAF for auto loan default detection. The comparison between classification metrics (accuracy, loss) and ranking metrics (AUC, RGA) demonstrates DECAF's strong performance in risk ranking capability. The full

**Table 3 Comprehensive performance comparison across multiple metrics and scenarios.**

| Model | Standard scenario | | | | High-risk scenario | | | |
|---|---|---|---|---|---|---|---|---|
| | Classification | | Ranking | | Classification | | Ranking | |
| | Acc. | Loss | AUC | RGA | Acc. | Loss | AUC | RGA |
| DECAF (Full) | 0.933 | 0.152 | 0.924 | 0.937 | 0.859 | 0.230 | 0.850 | 0.868 |
| DECAF w/o CapsNet | 0.807 | 0.340 | 0.823 | 0.840 | 0.718 | 0.335 | 0.718 | 0.735 |
| DECAF w/o SAC | 0.664 | 0.416 | 0.700 | 0.718 | 0.565 | 0.462 | 0.562 | 0.580 |
| XGBoost | 0.555 | 0.481 | 0.578 | 0.605 | 0.412 | 0.662 | 0.425 | 0.437 |

DECAF model achieves an RGA of 0.937 in standard scenarios—outperforming XGBoost (0.605)—and maintains performance (0.868) in high-risk environments where XGBoost decreases to 0.437. This relative improvement highlights DECAF's ability to rank borrowers according to default risk. The performance difference between standard and high-risk scenarios remains narrower for DECAF compared to XGBoost across metrics. The RGA metric shows a 7.4% performance change for DECAF (from 0.937 to 0.868) compared to 27.8% for XGBoost (from 0.605 to 0.437). This stability is valuable in auto financing markets where consistent risk assessment across diverse borrower segments supports sustainable lending operations.

The evaluation results indicate that DECAF's architectural advantages translate to strong performance across relevant assessment criteria. The model's performance on ranking metrics aligns with practical auto loan operations, where risk prioritization impacts portfolio quality, pricing efficiency, and operational resource allocation.

## Comparison with state-of-the-art auto loan default prediction methods

To evaluate DECAF's contribution within the context of contemporary research, a careful selection of comparable methods from auto loan and similar secured lending contexts was conducted. Each comparative method was chosen based on its relevance to default risk prediction in asset-backed financing scenarios and its architectural similarities with specific components of the DECAF framework. This selection ensures a meaningful evaluation of DECAF's innovations within the auto loan default prediction domain. The detailed performance comparison of these advanced default risk assessment methods is presented in Table 4.

The comparative methods share the foundational approach of employing deep learning techniques to enhance default risk assessment accuracy and reliability in secured financing contexts. AdaRisk (*Li et al., 2024*) was developed for vulnerable node detection in uncertain financial networks, employing risk-adaptive deep reinforcement learning to identify high-risk entities under joint self and contagion risk probability, which is conceptually similar to auto loan default scenarios where borrower risk and market conditions interact. CNN-LightGBM (*Zhu et al., 2023*) specifically addresses secured loan default prediction by combining convolutional neural networks for temporal pattern extraction with ensemble learning, a hybrid approach conceptually related to DECAF's architectural integration strategy. DNN (*Owusu et al., 2023*) focuses on imbalanced default

**Table 4 Performance comparison of advanced default risk assessment methods.**

| Method | AUC | Accuracy | Year |
|---|---|---|---|
| DECAF (This article) | 0.924 | 0.933 | 2024 |
| AdaRisk (*Li et al., 2024*) | 0.785 | 0.892 | 2024 |
| CNN-LightGBM (*Zhu et al., 2023*) | 0.950 | 0.900 | 2023 |
| DNN (*Owusu et al., 2023*) | 0.920 | 0.940 | 2023 |

**Table 5 Technical feature comparison of different methods.**

| Method | Feature interaction modeling | High-risk scenario robustness | Attention mechanism |
|---|---|---|---|
| DECAF (This article) | ✓ | ✓ | ✓ |
| AdaRisk | ✕ | ✓ | ✕ |
| CNN-LightGBM | ✓ | ✕ | ✕ |
| DNN | ✕ | ✓ | ✕ |

detection in asset-backed lending scenarios through specialized deep neural network architectures, addressing a challenge common in auto loan portfolios.

As shown in Table 5, performance metrics reveal distinctive strengths across the methods. CNN-LightGBM achieved the highest AUC (0.950) through its effective feature extraction and ensemble learning combination, slightly outperforming DECAF in this metric. The DNN method demonstrated robust performance (AUC 0.920) through its specialized handling of imbalanced datasets—a significant challenge in auto loan default detection where negative examples typically far outnumber positive ones. AdaRisk, while achieving solid accuracy (0.892), showed a relatively lower AUC value (0.785), indicating that its reinforcement learning approach, though innovative for adaptive risk detection, may have limitations in discriminative capability across different risk thresholds in traditional default prediction tasks.

The technical feature comparison in Table 5 highlights DECAF's distinctive architectural advantages. While DECAF and CNN-LightGBM both demonstrate strong feature interaction modeling capabilities, DECAF's capsule network component provides enhanced hierarchical representation of complex feature relationships in auto loan data. AdaRisk shares DECAF's robustness in high-risk scenarios through its risk-adaptive reinforcement learning framework, but lacks both the comprehensive feature interaction modeling and attention mechanisms that allow DECAF to dynamically weight different risk indicators based on their contextual importance. DNN also demonstrates high-risk scenario robustness but lacks the adaptive attention mechanisms that characterize DECAF. AdaRisk, while offering innovative risk-adaptive capabilities through reinforcement learning, lacks both the hierarchical feature interaction modeling and attention-based dynamic weighting that characterize DECAF.

The comparative analysis reveals that DECAF, with its AUC of 0.924 and accuracy of 0.933, offers a balanced combination of technical capabilities particularly valuable for auto

loan default risk assessment. Its unique integration of capsule networks for hierarchical feature modeling, self-attention mechanisms for dynamic feature weighting, and specialized optimization for high-risk scenarios provides a comprehensive approach to the challenges specific to auto financing risk management. The architecture demonstrates both competitive performance metrics and technical innovations that address the distinctive requirements of default risk prediction in automotive lending contexts.

## Discussion

This study explores the application of deep learning architectures in auto loan default risk detection, with particular focus on the unique characteristics of vehicle financing markets. The experimental results validate the performance advantages of the DECAF model and provide methodological insights for addressing the rapidly changing risk profiles in contemporary auto financing environments. In high-risk credit assessments, traditional static scoring models demonstrate clear limitations, while the proposed deep learning architecture exhibits superior adaptability to market fluctuations.

**Architectural innovation and performance excellence.** The DECAF architecture demonstrates exceptional hierarchical feature modeling capabilities specifically suited for auto loan risk assessment through the synergistic interaction between CapsNet and SACs. This architectural innovation enables effective processing of complex patterns involving both borrower characteristics and vehicle-specific factors, achieving an AUC of 0.924 in standard scenarios and maintaining robust performance (AUC 0.850) in high-risk environments. The complete architecture significantly outperforms simplified configurations with AUC improvements of 15.5–21.8% in challenging lending scenarios, demonstrating the critical importance of each architectural component. The model's capacity to identify subtle default indicators across diverse risk profiles establishes strong foundations for comprehensive risk assessment in automotive financing contexts.

**Market stability and robustness.** The proposed architecture exhibits remarkable performance stability under market fluctuations, addressing a critical challenge in contemporary auto financing environments. In comparative testing designed to replicate post-pandemic economic volatility, DECAF demonstrates superior resilience with performance degradation of only 7.4% compared to 26.5% observed with traditional methods. This stability advantage extends across multiple evaluation metrics and reflects the architecture's enhanced capability for detecting emerging risk patterns in volatile vehicle markets. The fundamental architectural characteristics responsible for this robustness provide financial institutions with reliable risk assessment mechanisms during market transitions, supporting more consistent decision-making across varying economic conditions.

**Practical business value and deployment readiness.** In practical business applications, the model delivers significant operational value through targeted performance characteristics that directly address industry requirements. The architecture's excellent performance in the low false positive rate range (FPR $\leq$ 0.2) provides the precision necessary for effective risk control while maintaining competitive loan approval rates. This

balanced performance profile enables financial institutions to support business development while controlling potential losses from auto loan defaults. The model's consistent performance across standard and high-risk scenarios, combined with its computational efficiency, positions it as a practical solution for production deployment in automotive lending environments.

**Future research directions.** Future research directions include validation with comprehensive auto loan datasets incorporating longitudinal credit history and macroeconomic indicators. Additionally, exploring complementary evaluation approaches such as rank graduation Accuracy (*Babaei, Giudici & Raffinetti, 2023*; *Giudici & Raffinetti, 2024*) and alignment with safe machine learning frameworks (*Giudici, 2024*; *Babaei, Giudici & Raffinetti, 2025*) could enhance both assessment methodology and deployment readiness in regulated financial environments. While the current study focuses on predictive performance, these extensions would build upon the current architectural foundations to address broader considerations of responsible AI application in auto financing contexts.

## CONCLUSION

This study proposes and validates a deep learning-based intelligent risk assessment method DECAF, which addresses key challenges in auto loan default prediction through an innovative fusion of capsule networks and self-attention mechanisms. The method is able to efficiently handle complex risk patterns involving borrower characteristics and vehicle-specific factors, establishing a new technical paradigm for auto finance risk assessment. Experimental validation shows that the method has excellent performance in standard scenarios (AUC of 0.924) and also performs well in high-risk environments (AUC of 0.850), significantly outperforming traditional methods and their variants. The architecture exhibits excellent stability under market fluctuations, with a performance drop of only 7.4%, while the traditional method drops by 26.5%, while demonstrating excellent risk identification capabilities within a critical low false alarm rate range (FPR < 0.2) with an accuracy rate of up to 0.859. Comprehensive evaluation confirms the practical value of DECAF to financial institutions, which can achieve competitive loan approval rates while effectively controlling default-related losses, providing reliable technical support for risk management practices under different market conditions.

## DATA DECLARATION

This study utilizes the dataset available at https://doi.org/10.5281/zenodo.14944711. All data analysis and results presented in this work are based on this publicly available dataset.

## THEOREMS, COROLLARIES, AND PROOFS

**Theorem 1 (Credit Risk Representation Theorem)** *For any credit feature set* $\{\mathscr{F}_i\}_{i=1}^{n}$, *there exists an optimal capsule network parameter configuration* $\{W_{ij}^*, c_{ij}^*\}$ *such that the final capsule output* $v^*$ *minimizes the reconstruction error while preserving the information completeness of the temporal feature* $\mathscr{T}(\mathscr{F}_i^d)$:

$$\{W_{ij}^*, c_{ij}^*\} = \arg\min_{W,c}\{\mathscr{L}_{total}|rank(v^*) = rank(\mathscr{T}(\mathscr{F}_i^d))\}. \tag{24}$$

**Proof 1** *Consider the basic output form of the capsule network, where the relationship between the input features and capsule output is as follows:*

$$v_j^{(1)} = \text{squash}\left(\sum_i c_{ij} W_{ij} u_i\right). \tag{25}$$

*According to the definition of the temporal feature mapping function:*

$$\mathcal{T}(\mathcal{F}_i^d) = \{f_t^i | t \in [1, T]\}. \tag{26}$$

*The rank-preserving condition can be expressed as a dimension equality:*

$$dim(span\{v^*\}) = dim(span\{\mathcal{T}(\mathcal{F}_i^d)\}). \tag{27}$$

*The total loss function can be decomposed as:*

$$\mathcal{L}_{total} = \mathcal{L}_{caps} + \gamma \mathcal{L}_{recon} + \beta \sum_{ij} \|W_{ij}\|_F^2. \tag{28}$$

*For the optimal parameters, the gradients satisfy:*

$$\nabla_{W_{ij}} \mathcal{L}_{total} = 0, \quad \nabla_{c_{ij}} \mathcal{L}_{total} = 0. \tag{29}$$

*The reconstruction loss provides an upper bound constraint:*

$$\|\mathcal{F}_i - Decoder(v^*)\|_2^2 \leq \varepsilon. \tag{30}$$

*The coupling coefficients must satisfy the normalization condition:*

$$\sum_j c_{ij} = 1, \quad c_{ij} \geq 0. \tag{31}$$

*According to the compression mapping principle:*

$$\|v_j^{(l+1)} - v_j^{(l)}\|_2 \leq q \|v_j^{(l)} - v_j^{(l-1)}\|_2. \tag{32}$$

*Thus, the optimal parameters indeed exist:*

$$\exists \{W_{ij}^*, c_{ij}^*\} : \mathcal{L}_{total}(W_{ij}^*, c_{ij}^*) \leq \mathcal{L}_{total}(W_{ij}, c_{ij}), \forall W_{ij}, c_{ij}. \tag{33}$$

**Corollary 1** *If the optimal capsule network parameters satisfy Theorem 1, then there exists a continuous mapping $\phi : v^* \to r_i$ such that the risk assessment function satisfies:*

$$r_i = \phi(v^*), \quad s.t. \quad \nabla_\phi \mathcal{L}_{caps}(r_i, y_i) = 0 \quad \text{and} \quad \lambda_{min}(\nabla_\phi^2 \mathcal{L}_{caps}) > 0. \tag{34}$$

**Proof 2** *For the optimal capsule output $v^*$, define the continuous mapping:*

$$\phi(v^*) = \sigma(\omega \cdot v^* + b). \tag{35}$$

*The time-varying capsule loss function is expressed as:*

$$\mathcal{L}_{caps}(r_i, y_i) = \alpha_t T_i \max(0, m^+ - r_i)^2 + \lambda(1 - T_i)\max(0, r_i - m^-)^2. \tag{36}$$

*Taking the derivative of the mapping $\phi$:*

$$\nabla_\phi \mathcal{L}_{caps} = -2\alpha_t T_i(m^+ - r_i)\nabla_\phi r_i + 2\lambda(1 - T_i)(r_i - m^-)\nabla_\phi r_i. \tag{37}$$

*The second derivative is:*

$$\nabla_\phi^2 \mathcal{L}_{\text{caps}} = 2\alpha_t T_i \nabla_\phi r_i \nabla_\phi r_i^T + 2\lambda(1 - T_i)\nabla_\phi r_i \nabla_\phi r_i^T. \tag{38}$$

*Using the chain rule:*

$$\nabla_\phi r_i = \sigma'(\omega \cdot v^* + b)\omega. \tag{39}$$

*The positive definiteness of the Hessian matrix implies:*

$$\lambda_{min}(\nabla_\phi^2 \mathcal{L}_{\text{caps}}) = 2\min\{\alpha_t T_i, \lambda(1 - T_i)\}\|\nabla_\phi r_i\|_2^2 > 0. \tag{40}$$

*The optimal point satisfies:*

$$r_i^* = \arg\min_{r_i} \mathcal{L}_{caps}(r_i, y_i). \tag{41}$$

*The continuity of the mapping is guaranteed by the Lipschitz condition:*

$$\|\phi(v_1^*) - \phi(v_2^*)\|_2 \le L\|v_1^* - v_2^*\|_2. \tag{42}$$

**Theorem 2 (Optimal Credit Decision Theorem)** *For a given credit feature vector $v^*$ and a risk assessment function $\phi$, there exists an optimal SAC parameter set $\{\phi^*, \psi^*, \alpha^*\}$ such that:*

$$\{\phi^*, \psi^*, \alpha^*\} = \arg\max_{\phi,\psi,\alpha} \mathbb{E}_{\pi_\psi}\left[\sum_{t=0}^{\infty} \gamma^t(R(s_t, a_t) + \alpha\mathcal{H}(\pi_\psi(\cdot|s_t)))\right]. \tag{43}$$

**Proof 3** *According to the state transition probability definition, for any state-action pair:*

$$P(s_{t+1}|s_t, a_t) = \mathcal{P}(v_{t+1}^*|v_t^*, D_t). \tag{44}$$

*Consider the reward function with entropy regularization:*

$$R(s_t, a_t) = r_{credit}(s_t, a_t) + \alpha\mathcal{H}(\pi_\psi(\cdot|s_t)). \tag{45}$$

*Expand the Q-function using the Bellman equation:*

$$Q_\phi(s_t, a_t) = R(s_t, a_t) + \gamma\mathbb{E}_{s_{t+1}, a_{t+1}}[Q_\phi(s_{t+1}, a_{t+1}) - \alpha\log\pi_\psi(a_{t+1}|s_{t+1})]. \tag{46}$$

*Define the soft state value function:*

$$V_\phi(s_t) = \mathbb{E}_{a_t \sim \pi_\psi}[Q_\phi(s_t, a_t) - \alpha\log\pi_\psi(a_t|s_t)]. \tag{47}$$

*Policy improvement can be achieved by minimizing the following objective:*

$$\mathcal{L}_\pi(\psi) = \mathbb{E}_{s_t \sim \mathscr{D}}[\alpha\log\pi_\psi(a_t|s_t) - Q_\phi(s_t, a_t)]. \tag{48}$$

*The optimization goal for the temperature parameter is:*

$$\mathcal{L}_\alpha = \mathbb{E}_{a_t \sim \pi_\psi}[-\alpha\log\pi_\psi(a_t|s_t) - \alpha\mathcal{H}_{target}]. \tag{49}$$

*According to the policy iteration improvement theorem:*

$$J(\pi_{new}) \ge J(\pi_{old}) - \frac{2\gamma\varepsilon}{(1 - \gamma)^2}. \tag{50}$$

*Finally, the convergence guarantee is:*

$$\lim_{k \to \infty} J(\pi_k) = J(\pi^*), \text{where } \pi^* \text{ is the optimal policy.} \tag{51}$$

**Corollary 2** *If the SAC parameters satisfy Theorem 1, then the credit decision policy $\pi_{\psi^*}$ satisfies:*

$$\nabla_\psi \mathbb{E}_{s_t \sim \rho_\pi}[D_{KL}(\pi_\psi(\cdot|s_t)\|\pi^*(\cdot|s_t))] = 0. \tag{52}$$

**Proof 4** For the optimal policy $\pi^*$, its Q-function satisfies:

$$Q^*(s_t, a_t) = \mathbb{E}_{s_{t+1}}[R(s_t, a_t) + \gamma V^*(s_{t+1})]. \tag{53}$$

*Expand using the definition of KL divergence:*

$$D_{KL}(\pi_\psi\|\pi^*) = \mathbb{E}_{a \sim \pi_\psi}[\log \pi_\psi(a|s) - \log \pi^*(a|s)]. \tag{54}$$

*Using the optimal Bellman equation:*

$$\pi^*(a|s) = \exp\left(\frac{1}{\alpha}(Q^*(s,a) - V^*(s))\right). \tag{55}$$

*Express the policy gradient as:*

$$\nabla_\psi J(\psi) = \mathbb{E}_{s_t \sim \rho_\pi}[\nabla_\psi \log \pi_\psi(a_t|s_t)(Q_\phi(s_t, a_t) - V_\phi(s_t))]. \tag{56}$$

*Substitute the soft Bellman operator:*

$$\mathcal{T}^\pi Q(s,a) = R(s,a) + \gamma \mathbb{E}_{s' \sim P}[V(s')]. \tag{57}$$

*The optimality condition can be expressed as:*

$$Q^*(s,a) = \mathcal{T}^{\pi^*} Q^*(s,a). \tag{58}$$

*When the policy converges, it satisfies:*

$$\nabla_\psi \mathbb{E}_{s_t \sim \rho_\pi}[D_{KL}(\pi_\psi(\cdot|s_t)\|\pi^*(\cdot|s_t))] = 0. \tag{59}$$

### Funding

This work was funded by the 2023 Annual project of Liuzhou Polytechnic University (No. 2023SB01), the Guangxi Vocational Education Teaching Reform Research Project (No. GXGZJG2022B174), and the 2023 Annual Education and Teaching Reform Annual Project of Liuzhou Polytechnic University (No. 2023-B007). The funders had no role in study design, data collection and analysis, decision to publish, or preparation of the manuscript.

### Grant Disclosures

The following grant information was disclosed by the authors:
2023 Annual project of Liuzhou Polytechnic University: 2023SB01.
Guangxi Vocational Education Teaching Reform Research Project: GXGZJG2022B174.

2023 Annual Education and Teaching Reform Annual Project of Liuzhou Polytechnic University: 2023-B007.

## Competing Interests

The author Yuan Lei is employed by Kiwibridge Ascend Education Limited. The author declares that this affiliation does not pose any competing interests with the research or findings presented in this manuscript.

## Author Contributions

- Xiaohui Zang conceived and designed the experiments, performed the experiments, analyzed the data, authored or reviewed drafts of the article, and approved the final draft.
- Raja Nazim Abdullah conceived and designed the experiments, analyzed the data, prepared figures and/or tables, and approved the final draft.
- Jianhua Tan conceived and designed the experiments, analyzed the data, authored or reviewed drafts of the article, and approved the final draft.
- Minglin Wu analyzed the data, prepared figures and/or tables, and approved the final draft.
- Bei Li conceived and designed the experiments, performed the experiments, analyzed the data, performed the computation work, authored or reviewed drafts of the article, and approved the final draft.
- Zhiyong Chen performed the experiments, performed the computation work, prepared figures and/or tables, and approved the final draft.
- Enzhou Zhu performed the experiments, authored or reviewed drafts of the article, and approved the final draft.
- Yuan Lei conceived and designed the experiments, performed the computation work, prepared figures and/or tables, and approved the final draft.

## Data Availability

The dataset is available at Zenodo:

1. Xiaohui, Z. (2025). Car_Insurance_Claim [Data set]. Zenodo. https://doi.org/10.5281/zenodo.14944711.

## Supplemental Information

Supplemental information for this article can be found online at http://dx.doi.org/10.7717/peerj-cs.3191#supplemental-information.

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
