# Peer review of "DECAF: deep capsule attention network for intelligent auto loan default risk detection"

_PeerJ Computer Science, doi:10.7717/peerj-cs.3191_

## Round 0.1 · original submission · Major Revisions

I have completed my evaluation of your manuscript. The reviewers recommend reconsideration of your manuscript following major revision. I invite you to resubmit your manuscript after addressing the comments below.

Reviewer 1 ·

Basic reporting

-

Experimental design

-

Validity of the findings

-

Additional comments

The authors submitted a deep learning model called DECAF. The research sounds promising, but several factors make this review impossible. The contribution of the work is considered low or insignificant compared with the current works in the research market. The presentation of the work is not well designed for journal publication. The results are not well presented to increase the contribution of the work.

Reviewer 2 ·

Basic reporting

This study introduces a novel deep learning architecture, DECAF, which integrates capsule networks with self-attention mechanisms to enhance auto loan default risk detection. By demonstrating robust performance across both conventional and high-risk lending scenarios, the model offers significant improvements in accuracy, false positive reduction, and resilience to market fluctuations.

However, given the proposed architectural enhancements and the claimed superiority over the baseline methods, some limitations remain, and the overall contribution remains not entirely convincing in terms of literature, novelty, and comparison. The literature reviewed is insufficient and lacks depth in terms of contextual relevance. Moreover, given the differences between the proposed model and the cited studies, a fair and meaningful comparison cannot be made. Many of the cited studies cover broader or different domains, such as risk analysis, credit platforms, fraud detection, or general credit risk assessment, and do not fully align with the specific objective of default risk estimation. In particular, some of these studies approach the problem primarily from a security perspective, rather than focusing on accurate and consistent default risk estimation.

Therefore, the literature review should be REVISED and expanded to include more directly relevant and supporting studies that are consistent with the main objective of the proposed approach.

Experimental design

The study utilizes the Car_Insurance_Claim dataset, which contains 19 feature dimensions, including 18 input features. However, the dataset appears to be insufficient in both scope and relevance for capturing the complexity of auto loan default risk modeling.

While the proposed DECAF architecture claims to effectively detect default risk in dynamic lending environments, the selected dataset lacks domain-specific richness and does not reflect the diverse, multi-source, and temporal nature of real-world auto loan data. Key variables typically associated with lending decisions, such as credit history, loan terms, borrower demographics, and financial behavior, are missing or not well-represented.

The study would benefit from applying the model to a more comprehensive and representative dataset.

Validity of the findings

The methodology section lacks formal consistency, which undermines the overall clarity and replicability of the proposed approach.

While the authors emphasise the superior performance and robustness of the model under market fluctuations, the comparative analysis with the underlying methods remains limited and not detailed enough. The experimental setup does not provide a transparent explanation of how the compared models were selected, tuned, or evaluated under equivalent conditions. Moreover, despite the claim of processing heterogeneous multi-source data, the use of a simple and limited dataset contradicts the methodological objective of modelling complex, real-world credit scenarios

·

Basic reporting

The paper proposes a deep learning model to predict and detect loan default. The proposal is interesting, but the proposed performance comparison metrics are a bit simple. The authors should improve the metrics by paying more attention to the existing literature. They should, in particular, employ a test-based model selection criterion, adopt the RGA (rank graduation accuracy) metrics, more general than AUC, as described in the following two papers:
Babaei, G., Giudici, P., Raffinetti, E. (2023). Explainable fintech lending. Journal of Economics and Business, 125-126, 106126.
Giudici, P., Raffinetti, E. (2024). RGA: a unified measure of predictive accuracy. Advances in Data Analysis and Classification

Experimental design

The authors should also link their work to the more comprehensive assessment framework known as safe machine learning, as described in:
Giudici, P. (2024). Safe machine learning. Statistics.
Babaei, Giudici, Raffinetti (2025). A Rank graduation box for SAFE AI. Expert systems with applications, 59, 125239.

Validity of the findings

The findings are interesting and should be strengthened using the previously described metrics.

---

## Round 0.2 · Minor Revisions

The review process is now complete. While finding your paper interesting and worthy of publication, the referees and I feel that more work could be done before the paper is published. My decision is therefore to provisionally accept your paper subject to minor revisions.

Reviewer 1 ·

Basic reporting

The authors have significantly improved the work by previous feedback and revisions. But still some revisions are necessary.

Experimental design

Some future possible test can be mentioned by authors. All the results figures are not readable in print, this must be resolved.

Validity of the findings

Some large missing spaces in theorems, this kills the space. Try to avoid itemize texts.

Additional comments

Authors must revise the references and avoid non-indexed publishers.

Reviewer 2 ·

Basic reporting

The authors have diligently incorporated the reviewers’ feedback, implementing extensive revisions that have notably improved the manuscript. While the current version demonstrates marked progress, additional minor refinements would further strengthen the paper.

The scope remains somewhat constrained, with limited coverage of relevant studies. Expanding the discussion to include a more comprehensive set of references would bolster the theoretical foundation.

Experimental design

Although the authors have made good-faith efforts to clarify the dataset’s characteristics, certain methodological details remain ambiguous. A more rigorous explanation—particularly regarding selection criteria, potential biases, and preprocessing steps—would enhance reproducibility.

Validity of the findings

While acknowledging dataset constraints is necessary, overemphasizing them as a task for future studies risks understating the present work’s contributions. A balanced framing, focusing on both current insights and tractable extensions, would be more effective.

·

Basic reporting

The authors took my suggestions into account. the paper can be accepted

Experimental design

good

Validity of the findings

good

Additional comments

good

---

## Round 0.3 · accepted · Accept

We are happy to inform you that your manuscript has been accepted for publication since the comments have been addressed.

Reviewer 1 ·

Basic reporting

Authors have met with previous comments. The text and contribution of the work is much clear now.

Experimental design

The authors covered the figure issues and make them readable. avoid 7.4\% in abstract. authors must revise these errors.

Validity of the findings

The experimental findings are revised.

Reviewer 2 ·

Basic reporting

Thanks for the revised version. I am satisfied that the authors have addressed the issues raised during my first review. They responded to the comments and suggestions for changes included in the attached annotated manuscript, and they also revised different parts of the paper according to my feedback in the review. Best wishes for this publication.

Experimental design

Thanks for the revised version. I am satisfied that the authors have addressed the issues raised during my first review. They responded to the comments and suggestions for changes included in the attached annotated manuscript, and they also revised different parts of the paper according to my feedback in the review. Best wishes for this publication.

Validity of the findings

Thanks for the revised version. I am satisfied that the authors have addressed the issues raised during my first review. They responded to the comments and suggestions for changes included in the attached annotated manuscript, and they also revised different parts of the paper according to my feedback in the review. Best wishes for this publication.